cognition

linguistic memory, information theory, Fermi estimation

**Author for correspondence:**
Francis Mollica
e-mail: mollicaf@gmail.com

# Humans store about 1.5 megabytes of information during language acquisition

Francis Mollica[1] and Steven T. Piantadosi[2]

[1]Department of Brain and Cognitive Sciences, University of Rochester, Rochester, NY, USA
[2]Department of Psychology, University of California, Berkeley, CA, USA

 FM, 0000-0003-1008-5397

We introduce theory-neutral estimates of the amount of information learners possess about how language works. We provide estimates at several levels of linguistic analysis: phonemes, wordforms, lexical semantics, word frequency and syntax. Our best guess is that the average English-speaking adult has learned 12.5 million bits of information, the majority of which is lexical semantics. Interestingly, very little of this information is syntactic, even in our upper bound analyses. Generally, our results suggest that learners possess remarkable inferential mechanisms capable of extracting, on average, nearly 2000 bits of information about how language works *each day* for 18 years.

## 1. Introduction

One of the foundational debates about human language centres on an issue of scale: is the amount of information about language that is learned substantial (empiricism) or minimal (nativism)? Despite theoretical debates on how much of language is or can be learned, the general question of the amount of information that *must* be learned has yet to be quantified. Here, we provide an estimate of the amount of information learners must extract in the process of language acquisition. We provide *lower-bound*, *best guess*, and *upper-bound* estimates of this information, using a 'back of the envelope' approach that is popular in physics. During the testing of the atomic bomb, physicist Enrico Fermi famously estimated the strength of the bomb by dropping scraps of paper as the blast passed. He noted that they were blown about 8 feet by the explosion and, after quickly computing in his head, announced that the blast was equivalent to about 10 000 tons of TNT [1]. The true answer was 18 000 tons, meaning Fermi's crude experiment and quick calculation got the right answer to within a factor of two. Similar back-of-the-envelope *Fermi* calculations are commonly used in physics as a sanity check on theories and

computations.[1] However, such sanity checks are needed—although rarely applied—in fields that suffer from under-constrained theories, like psychology.

We apply this approach of rough estimation in order to quantify a lower bound on the number of *bits per day* that language learners must extract and remember from their environments. While a substantial amount of literature has focused on the differences between nativist and empiricist approaches, when they are translated into the domain of information theory, nativism and empiricism may blur together. Specifically, hard nativist constraints on sets of hypotheses may not necessarily provide much more information than the correct set of biases over an unconstrained space. Since we do not know much about which initial language learning biases children have, and how they interact with cognitive constraints, theories about how the hypotheses are constrained (or not) do not unambiguously determine the number of bits learners must store.

So, instead of debating nativism versus empiricism, we take up the challenge of quantifying how much information on language must be learned in order to inform the underlying acquisition theories. To avoid dependence on a particular representation scheme, we focus on the possible *outcomes* of learning a language, i.e. we compute the number of bits required to specify the target outcome from a plausible space of logically possible alternatives. To avoid dependence on a particular learning algorithm, we study the problem abstractly without reference to *how* learning works, but rather based on how much a relatively unbiased (e.g. maximum entropy) learner would have to store.

Our study is inspired by prior work which has aimed to characterize the capacity of human memory. Early literature approached the question of memory capacity from a neuroanatomical perspective. Upper bounds on memory capacity have been estimated via the number of synapses in cortex ($10^{13}$ bits) or the number of impulses conducted in the brain across a lifetime ($10^{20}$ bits) [2]. More recently, bounds for information transfer in neural coding have been estimated using information theoretic techniques [3,4]. Working from behavioural performance, Landauer [5] used a variety of techniques to estimate the number of bits of information humans must have in memory in order to show a given level of task performance. In one example, he converted accuracy in a recognition memory task to bits by imagining that each image was stored using a random code. This technique has been used recently by Brady *et al.* [6] in a large-scale empirical study, which estimated that human memory could encode $2^{13.8}$ unique items. Even more recently, [7] estimated 4- and 6-year-old children's memory capacity to be $2^{10.43}$ unique items. Strikingly, both of these estimates lie within Landauer's estimated range of 10–14 bits per item. Landauer also used a dictionary study to estimate the number of words that Stanford students knew, and converted the estimates for a phonetic code into bits, requiring about 30–50 bits per word [5]. All of his estimates converged on the same order of magnitude, suggesting that the 'functional capacity' of human memory is in the order of $10^9$ bits. A detailed critique of Landauer can be found in [8], with a response given by Landauer [9].

Our focus here is on estimating *linguistic* knowledge across multiple levels of structure and function: phonemes, wordforms, lexical semantics, word frequency and syntax. At each level, there is a large space of logically possible linguistic representations (e.g. acoustic cue values, syntactic parses). The challenge for learners is to discover and store which representations are used in their language. Tools in information theory allow us to estimate the relevant quantities. First, we assume that before learning, children begin with a certain amount of uncertainty over the required representation, $R$, denoted as $\mathbf{H}[R]$. Shannon entropy [10] quantifies the number of bits that must be received on average to remove uncertainty about what $R$ is the true one. After observing some data $D$, learners will have a new amount of uncertainty (perhaps zero) over $R$, denoted $\mathbf{H}[R|D]$. Note that here, $D$ is not a random variable, but rather a specific value of data in learning a given language.

We can formalize the amount of information that $D$ provides about $R$, here denoted as $\mathbf{\Delta H}[R|D]$ as the difference between $\mathbf{H}[R]$ and $\mathbf{H}[R|D]$:

$$\mathbf{\Delta H}[R|D] = \mathbf{H}[R] - \mathbf{H}[R|D] = -\sum_{r \in R} P(r) \log P(r) + \sum_{r \in R} P(r|D) \log P(r|D). \qquad (1.1)$$

This quantity, i.e. the reduction in entropy, gives the amount of information that $D$ (e.g. data from learning) provides about a representation $R$.[2] Thus, in order to estimate the amount of information learners must have acquired, it suffices to estimate their uncertainty before learning, $\mathbf{H}[R]$, and

---

[1]These computations are also used as a training exercise that allows surprising quantities to be approximated. An example is to compute the thickness of a car tire that is worn off with each rotation. Here is a hint: you can use your knowledge of how many miles car tires last for and how much thickness they lose over their lifetime.

[2]The average of $\mathbf{\Delta H}[R|D]$ over $D$ is the *mutual information* between $R$ and $D$ [11].

subtract from it their uncertainty after learning $\mathbf{H}[R|D]$. The resulting value will tell us the number of bits of information that the learning data $D$ has provided.[3]

# 2. Results

We will build up our estimates separately for each linguistic domain. The results of each section are summarized in table 1. Electronic supplementary material, table S1 summarizes the key assumptions behind each of our estimations.

## 2.1. Information about phonemes

Our phonemic knowledge enables us to perceive discrete linguistically-relevant sounds, or phonemes, from rich high-dimensional but noisy speech signals. Before a child knows the sounds of their language, they have uncertainty over the acoustic features of speech sounds. After learning their language, children have much less uncertainty over the acoustic features of speech sounds as they now have several acoustic cues to help them identify phonemes. Following the above logic, the decrease in the amount of uncertainty children have about where their speech sounds lie in acoustic space is the amount of information they must now store about phoneme cues.

Identifying linguistically relevant acoustic cues has proven challenging for scientists, as there is no obvious invariance, or uniquely identifying component, in speech sounds. For our estimation, we analyse the information stored for three well-studied acoustic cues: voice onset time (VOT) in ms—a cue to voiced-voiceless distinctions (e.g. the difference between /p/ and /b/), central frication frequency in barks—a cue to the place of articulation for fricatives (e.g. the difference between /f/ and /s/), and the first two formant frequencies of vowels in mels—a cue to vowel identity. We assume that initially, learners have maximum uncertainty along each cue $R$, following uniform distributions bounded by the limits of perception. In this case, each $r \in R$ has an equal probability of $P(r) = 1/(B - A)$, giving

$$\mathbf{H}[R] = -\int P(r) \log P(r)\, dr = \log(B - A), \qquad (2.1)$$

where $B$ and $A$ are, respectively, the upper and lower bounds of perception. For VOT, we assume the range is $-200$ to $200$ ms. For frequencies, we assume bounds on human hearing of $20$–$20\,000$ Hz, which translate to $0.2$–$24.6$ in barks and $32$–$3817$ in mels. As a measure of the uncertainty over the cue dimension after learning, we will assume that the speaker's knowledge is captured by a normal prior distribution, giving $\mathbf{H}[R|D]$ as

$$\mathbf{H}[R|D] = \int N(x|\mu, \sigma) \log N(x|\mu, \sigma)\, dx = \frac{1}{2} \log(2\pi e \sigma^2), \qquad (2.2)$$

where $N$ is a normal probability density function,[4] and $\mu$ and $\sigma$ are the participants' inferred mean and standard deviation. To find $\sigma$ for real humans, we use the values inferred by [12, table 7] to account for the perceptual magnet effect.[5]

We find that language users store 3 bits of information for voiceless VOT, 5 bits for voiced VOT, 3 bits for central frication frequency and 15 bits for formant frequencies. As these acoustic cues are only a subset of the cues required to identify consonant phonemes, we assume that consonants require three cues; each cue requiring 5 bits of information. For vowels, we do not adjust the 15 bits of information conveyed by formant frequencies. As a best guess, again paying attention primarily to the order of magnitude, we assume there are 50 phonemes each requiring 15 bits, totalling 750 bits of information. For lower and upper estimates, we introduce a factor of two error [375–1500 bits].

## 2.2. Information about wordforms

When dealing with wordforms, the first challenge is to define a 'word', a term which could be used to refer to lemmas, phonological forms, families of words, senses, etc. Entire dissertations could be (and

---

[3]In the case of continuous distributions, (1.1) has continuous analogues where the sums turn into integrals.

[4]Using a normal distribution with the domain truncated to our perceptual bounds does not change our estimate.

[5]For vowels, we extend these distributions to their multi-dimensional counterparts as formant space is (at least) two dimensional.

**Table 1.** Summary of estimated bounds across levels of linguistic analysis.

| section | domain | lower bound | best guess | upper bound |
|---|---|---|---|---|
| 2.1 | phonemes | 375 | 750 | 1500 |
| 2.2 | phonemic wordforms | 200 000 | 400 000 | 640 000 |
| 2.3 | lexical semantics | 553 809 | 12 000 000 | 40 000 000 |
| 2.4 | word frequency | 40 000 | 80 000 | 120 000 |
| 2.5 | syntax | 134 | 697 | 1394 |
| | total (bits) | 794 318 | 12 481 447 | 40 762 894 |
| | total per day (bits)[a] | 121 | 1900 | 6204 |

[a]For this value, we assume language is learned in 18 years of 365 days.

have been) written on these distinctions. These difficulties are in part why the Fermi approach is so useful: we do not need to make strong theoretical commitments in order to study the problem if we focus on rough estimation of orders of magnitude. Estimates of the number of words children acquire range in the order of 20 000–80 000 total wordforms [13]. However, when words are grouped into families (e.g. 'dog' and 'dogs' are not counted separately) the number known by a typical college student is more in the range of 12 000–17 000 [14,15] (although see [16] for an estimate over twice that size). Lexical knowledge extends beyond words, too. Jackendoff [17] estimates that the average adult understands 25 000 idioms, items out of the view of most vocabulary studies. Our estimates of *capacity* could, of course, be based on upper bounds on what people *could* learn, which, to our knowledge, have not been found. Looking generally at these varied numbers, we will use an estimate of 40 000 as the number of essentially unique words/idioms in a typical lexicon.

The most basic thing each learner must acquire about a word is its phonemic wordform, meaning the sequence of phonemes that make up its phonetic realization. If we assume that word forms are essentially memorized, then the entropy $\mathbf{H}[R|D]$ is zero after learning—e.g. for all or almost all words, learners have no uncertainty of the form of the word once it has been learned. The challenge then is to estimate what $\mathbf{H}[R]$ is: before learning anything, what uncertainty should learners have? To answer this, we can note that $\mathbf{H}[R]$ in (1.1) can be viewed as an *average* of the negative log probability of a wordform, or $-\log P(R)$. Here, we use a language model to estimate the average negative log probability of the letter sequences that make up words and view this as an estimate of the amount of entropy that has been removed for *each* word. In other words, the average surprisal of a word under a language model provides one way to estimate the amount of uncertainty that learners who know a given word must have removed.[6,7]

To estimate these surprisals, we used the CELEX database [18]. We computed the surprisal of each word under 1-phone, 2-phone, 3-phone and 4-phone models (see [19]) trained on the lexicon. This analysis revealed that 43 bits per word on average are required under the 1-phone, 33 bits per word under the 2-phone, 24 under the 3-phone and 16 under the 4-phone model. Noting the sharply decreasing trend, we will assume a lower bound of about 5 bits per word to store the phonetic sequence, a 'best guess' of 10 bits per word and an upper bound of 16 as in the 4-phone.[8] When our best guess is multiplied by the size of the lexicon (40 000 words), that gives an estimate of 400 000 [200 000–640 000] bits of lexical knowledge about the phonetic sequences in words.

## 2.3. Information about lexical semantics

The information contained in lexical semantics is difficult to evaluate because there are no accepted theories of semantic content, or conceptual content more generally [20]. However, following Fermi, we

[6]In this view, we neglect the complexity of the language model, which should be a much smaller order of magnitude than the number of bits required for the entire lexicon.

[7]Analogously, we can view the surprisal as the number of bits that must be remembered or encoded for a particular outcome—e.g. to learn a specific wordform.

[8]As the average word length in this database is approximately 7.5 phonemes, this gives slightly over 1 bit per phoneme.

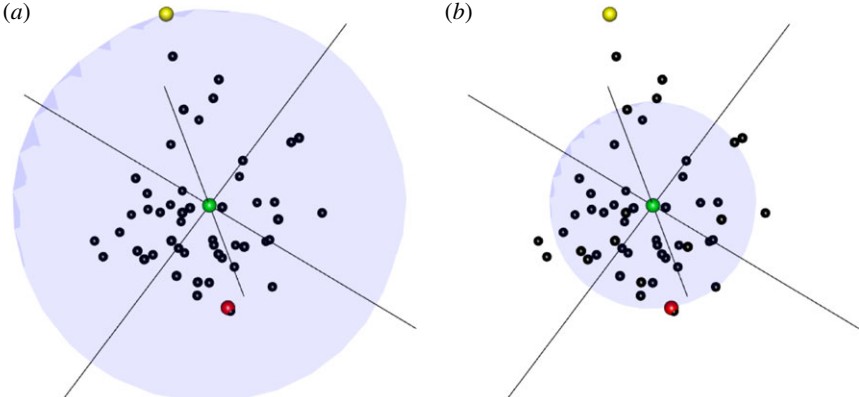

**Figure 1.** The shaded spheres represent uncertainty in semantic space centred around a word (in green). (*a*) The uncertainty is given with respect to the word's farthest connection in semantic space (in yellow), representing *R*. (*b*) The uncertainty is given with respect to the *N*th nearest neighbour of the word (in red), representing *r*. The reduction in uncertainty from *R* to *r* reflects the amount of semantic information conveyed by the green word.

can make very simplified assumptions and try to estimate the general magnitude of semantic content. One way to do this is to imagine that the set of word meanings are distributions in an *N*-dimensional semantic space. If we assume that the entire space is a Gaussian with standard deviation *R* and the standard deviation of an individual word meaning is *r*, then we can compute the information contained in a word meaning as the difference in uncertainty between an *N*-dimensional Gaussian with radius *R* when compared with one with radius *r*. The general logic is shown in figure 1. The 'space' shown here represents the space of semantic meanings, and words are viewed as small distributions in this space covering the set of things in the extension of the word's meaning. Thus, when a leaner acquires a word like 'accordion', they know that the word refers to some relatively small subset (of size *r*) of possible objects, but they may not be certain on the details (Does the extension cover harmoniums? Concertinas? Bayans?). The reduction in entropy from a total semantic space of size *R*—no idea what a word means—to one of size *r* is what we use to approximate the amount of information that has been learned.

Equation (2.2) above gives the change in entropy for a one-dimensional Gaussian. However, the dimensionality of semantic space is considerably larger. In the case of an *N*-dimensional Gaussian, with independent dimensions and (constant, or homogeneous) standard deviation $\sigma$ in each dimension, the entropy is given by

$$\mathbf{H}[R] = \frac{N}{2}(1 + \log 2\pi + \log \sigma). \tag{2.3}$$

This means that if we go from an *R* standard deviation Gaussian to an *r* standard deviation one, the amount of information we have learned is the difference between these entropies,

$$\Delta\mathbf{H}[R|D] = \frac{N}{2}(1 + \log 2\pi + \log R) - \frac{N}{2}(1 + \log 2\pi + \log r) = \frac{N}{2}\log\frac{R}{r}. \tag{2.4}$$

We estimate *R* and *r* in several different ways by looking at WordNet [21] to determine the closeness of each word to its neighbours in semantic space. In particular, we take *r* to be a characteristic distance to nearby neighbours (e.g. the closest neighbours), and *R* to be a characteristic distance to far away ones (e.g. the max distance). Note, this assumes that the size of a Gaussian for a word is about the same size as its distance to a neighbour, and in reality this may underestimate the information a word meaning contains because words could be much more precise than their closest semantic neighbour.

Figure 2 shows $\frac{1}{2}\log(R/r)$ for several estimates of *R* and *r* for 10 000 random nouns in WordNet. The likely values fall within the range of 0.5–2.0 bits. Because we are plotting $\frac{1}{2}\log(R/r)$ and not $(N/2)\log(R/r)$, these values may be interpreted as the number of *bits per dimension* that lexical semantics requires. For instance, if semantic space was one-dimensional then it would require 0.5–2.0 bits per word; if semantic space were 100-dimensional, lexical semantics would require 50–200 bits per word. The nearness of these values to 1 means that even continuous semantic dimensions can be viewed as *approximately* binary in terms of the amount of information they provide about meaning.

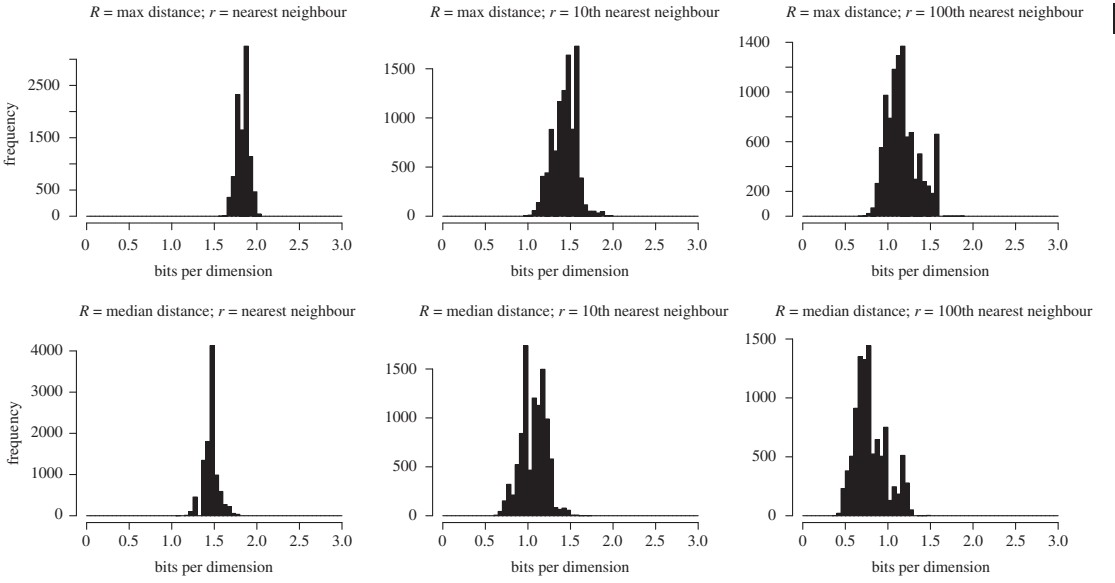

**Figure 2.** Histograms showing the number of bits-per-dimension ($\frac{1}{2}\log(R/r)$) for various estimates of $R$ and $r$. These robustly show that $0.5 - 2.0$ bits are required to capture semantic distances.

The dimensionality of semantic space has been studied by [22,23], with numbers ranging from 100 to 500 dimensions. Our best guess will use 1 bit per dimension and 300 dimensions following [22] for 12 000 000 bits. Our upper bound uses 2 bits-per-dimension and 500 dimensions for a total of 40 000 000 bits.

For our lower bound in this domain, we may pursue a completely alternative technique, which surprisingly, gives a similar order of magnitude as our best guess. If there are 40 000 lexical items that must be learned, we can assume that they correspond to 40 000 distinct concepts (à la *principle of contrast* [24]). In the 'most nativist' case, favoured by Fodor [25], we could assume that there are a corresponding 40 000 meanings for these words that learners innately have. In this case, the problem of learning is figuring out which of the 40 000! pairings of words and concepts is the correct one. It will take $\log_2(40\,000!) \approx 553\,809$ bits of information to specify which of these is correct. We will use this as our lower bound. While this seems like an unmanageable task for the child, it is useful to imagine how much information is conveyed by a single pedagogical learning instance. Our estimate is derived by a combinatorial argument: to choose the first word's meaning, there are $N$ choices, for the second there are $N-1$, and so on. The total number of choices is therefore $N \cdot (N-1) \cdot (N-2) \cdot (N-40\,000) = N!/(N-40\,000)!$. So if initially $N = 400\,000$ (553 809 bits), there will be $N = 39\,999$ (553 794 bits) after one correct mapping is learned, meaning that a single pedagogical instance rules out 39 999 possible pairings or, equivalently, conveys 15.29 bits.

## 2.4. Information about word frequency

Word frequencies are commonly studied in psychology as factors influencing language processing and acquisition (e.g. [26–31]) as well as for their peculiar Zipfian distribution [32]. However, relatively little work has examined the fidelity of people's representation of word frequency, which is what is required in order to estimate how much people know about them. In one extreme, language users might store perhaps only a single bit about word frequency, essentially allowing them to categorize high- versus low-frequency words along a median split. On the other extreme, language users may store information about word frequency with higher fidelity—for instance, 10 bits would allow them to distinguish $2^{10}$ distinct levels of word frequency as a kind of psychological floating point number. Or, perhaps language learners store a full ranking of all 40 000 words in terms of frequency, requiring $\log(40\,000!) = 553\,809$ bits of information.

In an experimental study, we asked participants from Mechanical Turk ($N = 251$) to make a two-alternative forced choice to decide which of two words had higher frequency.[9] Words were sampled from the lexical database SUBTLEX [33] in 20 bins of varying log frequency. We removed words

[9]Participants were instructed that we were interested in their first impression and that there was no need to look up word frequencies.

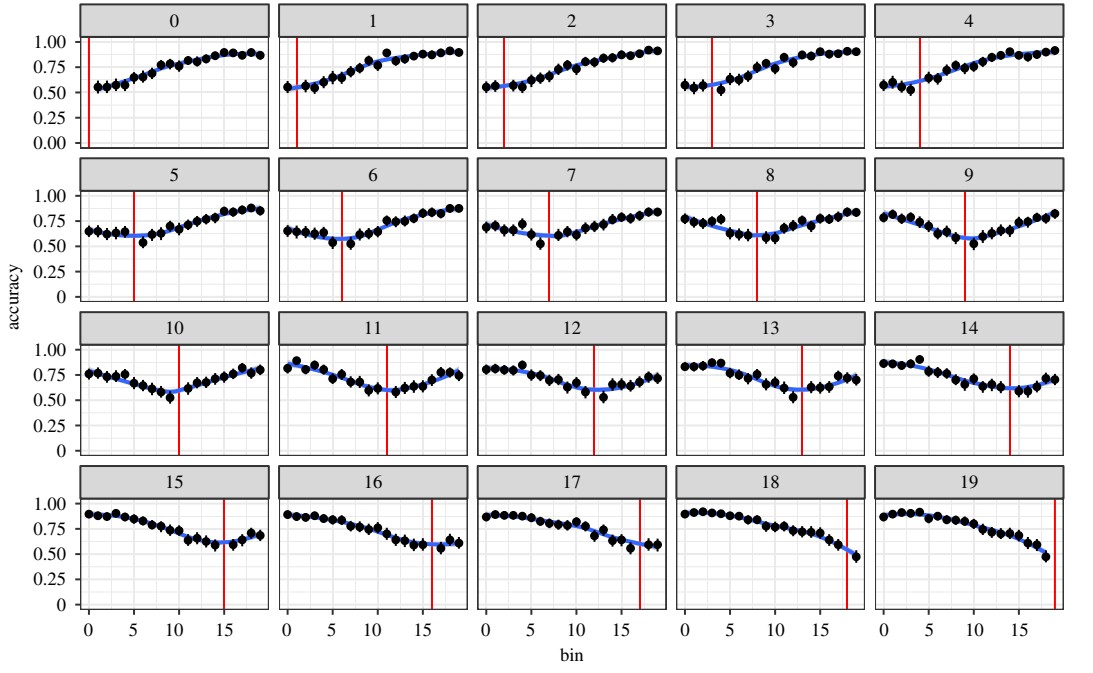

**Figure 3.** Accuracy in frequency discrimination accuracy as a function of log word frequency bin faceted by log reference word frequency bin. Vertical red lines denote within bin comparison. Line ranges reflect 95% bootstrapped confidence intervals.

below the bottom 30th percentile (frequency count of 1) and words above the upper 99th percentile in word frequency in order to study the intermediate-frequency majority of the lexicon. Each participant completed 190 trials.

Participants' accuracy in answering is shown in figure 3. The $i$th subplot shows participants' accuracy ($y$-axis) in distinguishing the $i$th bin from each other $j$th bin, with the red line corresponding to $i = j$. This shows, for instance, that people are poor at distinguishing very close $i$ and $j$ (near the red line), as should be expected.

Participants' overall accuracy in this task was 76.6%. Neglecting the relatively small difference in accuracy (and thus fidelity) with a word's absolute frequency, this accuracy can be modelled by imagining that participants store $M$ levels of word frequencies. Their error rate on this task will then be given by the probability that two words fall into the same bin, or $1/M$. Setting $1/M = 1 - 0.766$ gives $M \approx 4$, meaning that participants appear to track approximately four categories of frequencies (e.g. high, medium-high, medium-low, low). This trend can also be observed in figure 3, where the flat bottom of the trough in each plot is approximately 5 bins wide, meaning that each bin cannot be well distinguished from its five nearest neighbours, giving a total effective number of bins for participants as $20/5 = 4$.

If $M = 4$, then participants would only need to learn $\log 4 = 2$ bits of information about a word's frequency, as a best guess. This would yield a total of $2 \times 40\,000 = 80\,000$ bits total across the entire lexicon. We construct our lower and upper bounds by introducing a factor of two error on the computation (e.g. per word lower bound is 1 bit and upper is 3 bits). It is important to note that by assuming objective frequency rankings, our estimate is conservative. If we could analyse participants' responses with regard to their subjective frequency rankings, we would expect to see greater accuracy reflecting higher resolution representations of frequency.

## 2.5. Information about syntax

Syntax has traditionally been the battleground for debates about how much information is built-in versus learned. Indeed, syntactic theories span the gamut from those that formalize a few dozen binary parameters [34,35] to ones that consider alternative spaces of infinite models (e.g. [36,37]) or data-driven discovery from the set of all parse trees [38]. In the face of massively incompatible and experimentally under-determined syntactic theories, we aim here to study the question in a way that is as independent as possible from the specific syntactic formalism.

We do this by noting that every ordinary English speaker's knowledge of syntax provides enough information to correctly parse every sentence of English. In many cases, the sentences of English will share syntactic structure. However, we can imagine a set $s_1, s_2, \ldots, s_n$ of sentences which share as little syntactic structure as possible between each $s_i$ and $s_j$. For instance,

$$\text{Bill [met John].} \qquad \text{[Jill's sister] cried.} \tag{2.5}$$

both have three words but have largely non-overlapping syntactic structures due to the use of a transitive verb in the first and a possessive and intransitive verb in the second. We will call these 'essentially independent' sentences when they share almost no syntactic structure. In this case, the bits specifying these parses can be added together to estimate the total information learners know. If the sentences were not essentially independent in terms of syntactic structure, information from one sentence would tell us how to parse information from another, and so adding together the information for each would be an overestimate of learners' knowledge.

We assume that learners who do not know anything about a parse of a sentence $s_i$ start with a maximum entropy distribution over each parse, assigning to each an equal probability of one over the number of logically possible parses of $s_i$, so that

$$\mathbf{H}[R] = -\sum_{r \in R} \frac{1}{\#parses} \log \frac{1}{\#parses} = \log(\#parses). \tag{2.6}$$

We assume the knowledge of an adult leaves zero uncertainty in general, yielding $\mathbf{H}[R|D] = 0$ so that

$$\mathbf{\Delta H}[R|D] = \mathbf{H}[R] - \mathbf{H}[R|D] = \log(\#parses) \tag{2.7}$$

for a single sentence $s_i$. In general, the number of logically possible parses can be computed as the number of binary trees over $s_i$, which is determined only by the length of $s_i$. The $(l-1)$th Catalan number gives the number of possible binary parses for a sentence of length $l$. Then, the number of bits required to specify *which* of these parses is correct is given by $\log C_{l-1}$. The Catalan numbers are defined by

$$C_n = \frac{1}{n+1}\binom{2n}{n}. \tag{2.8}$$

As an example, to determine each of (2.5), knowledge of syntax would have to specify $\log C_2 = 1$ bit, essentially specifying whether the structure is $((\circ \circ) \circ)$ or $(\circ (\circ \circ))$. But $C_n$ grows exponentially—for instance, $C_{10} = 16796$, requiring 14 bits to specify which parse is correct for an 11-word sentence.

Looking at a collection of sentences, if $s_i$ has length $l(s_i)$, then the total amount of information provided by syntax will be given by

$$\sum_{i=1}^{n} \log C_{l(s_i)-1}. \tag{2.9}$$

Again, (2.9) assumes that there is no syntactic structure shared between the $s_i$—otherwise (2.9) overestimates the information by failing to take into account the fact that some bits of information about syntax will inform the parses of distinct sentences. Our upper and lower bounds will take into account uncertainty about the number of distinct sentences $s_i$ that can be found.

To estimate the number of such sentences, we use the textbook linguistic examples studied by [39]. They present 111 sentences that are meant to span the range of interesting linguistic phenomena and were presented independently in [40]. Our best estimate is therefore (2.9) taking $s_i$ to be the lengths of these sentences. We take the lower bound to be (2.9) where $l(s_i)$ is *half* the sentence length of $s_i$, meaning that we assume that only half of the words in the sentence participate in a structure that is independent from other sentences. For an upper bound, we consider the possibility that the sentences in [39] may not cover the majority of syntactic structures, particularly when compared to more exhaustive grammars like [41]. The upper bound is constructed by imagining that linguists could perhaps construct two times as many sentences with unique structures, meaning that we should double our best guess estimate. Notably, these tactics to bound the estimate do not qualitatively change its size: human language requires very little information about syntax—697 [134–1394] bits. In either case, the number is much smaller than most other domains.

# 3. Discussion

Summing across our estimates for the amount of information language users store about phonemes, wordforms, lexical semantics, word frequency and syntax, our best guess and upper bound are on the order of 10 million bits of information, the same order as [5]'s estimate for language knowledge. It may seem surprising but, in terms of digital media storage, our knowledge of language almost fits compactly on a floppy disk. The best-guess estimate implies that learners must be remembering 1000–2000 bits *per day* about their native language, which is a remarkable feat of cognition. Our lower bound is around a million bits, which implies that learners would remember around 120 bits each day from birth to 18 years. To put our lower estimate in perspective, each day for 18 years a child must wake up and remember, perfectly and for the rest of their life, an amount of information equivalent to the information in this sequence,

$$0110100001101001011001000110010001100101011011100110000101100011$$
$$0110001101101111011100100110010001101001011011110110110$$

Naturally, the information will be encoded in a different format—presumably one which is more amenable to the working of human memory. But in our view, both the lower and best-guess bounds are explainable only under the assumption that language is grounded in remarkably sophisticated mechanisms for learning, memory, and inference.

There are several limitations to our methods, which is part of the reason we focus on orders of magnitude rather than precise estimates. First, our estimates are rough and require simplifying assumptions (listed in electronic supplementary material, table S1). Second, there are several domains of linguistic knowledge whose information content we do not estimate here including word predictability, pragmatic knowledge, knowledge of discourse relations, prosodic knowledge, models of individual speakers and accents, among others. Many of these domains are difficult because the spaces of underlying representations are not sufficiently well formalized to compute information gains (e.g. in pragmatics or discourse relations). In other areas like people's knowledge of probable sequences of words, the information required is difficult to estimate because the same content can be shared between constructions or domains of knowledge (e.g. the knowledge that 'Mary walks' and 'John walks' are high probability may not be independent from each other, or from knowledge about the lexical semantics of the verb). We leave the estimation of the amount of information language users store about these domains of language to further research.

Importantly, our estimates vary on orders of magnitude across levels of representation. These differences could suggest fundamental differences in the learning mechanism for specific language learning problems. As these analyses show, the majority of information humans store about language is linked to words, specifically lexical semantics, as opposed to other systems of language knowledge, such as phonemes and syntax. In fact, the estimate for syntax is of a similar order of magnitude proposed by some nativist accounts, in that the number of bits required for syntax is in the hundreds, not tens of thousands or millions. To illustrate, if syntax learning is principally completed in the first 5 years, children would have to learn a single bit about syntax every 2–3 days on average. Despite this, the possible outcomes for learners in our best guess for syntax consists of $2^{697} \approx 10^{210}$ different systems. In other words, learners would still need the ability to navigate an immense space of possibilities, far greater than the number of atoms in the universe (approx. $10^{80}$). In the other areas of language, even more enormous hypothesis spaces are faced as well, pointing to the existence of powerful inferential mechanisms.

Turning back to nativism and empiricism, it is unfortunate that the majority of the learnability debates have centred on syntactic development, which requires far less information in total than even just a few word meanings. Despite the remarkable mechanisms that must be deployed to learn hundreds of thousands or millions of bits about lexical semantics, there are *no viable accounts* of lexical semantics representation and learning, either from empiricists or nativists (despite some claims by Fodor [25]). Our results suggest that if any language-specific knowledge is innate, it is most likely for helping tackle the immense challenge of learning lexical semantics, rather than other domains with learnability problems that require orders of magnitude less information.

**Ethics.** This project was approved by the University of Rochester Research Subjects Review Board (case no. RSRB00053982). Informed consent was obtained for each participant in our experiment.

**Data accessibility.** All data and code can be found in [42].

Authors' contributions. The authors contributed equally to the design, coding and writing of this manuscript. All authors gave final approval for submission.

Competing interests. We declare we have no competing interests.

Funding. We received no funding for this study.

Acknowledgements. The authors thank Daniel Gildea for helpful discussions and Evelina Fedorenko for comments on early drafts.

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
