## [Reviewer comments · Royal Society Open Science]

Review History

RSOS-181393.R0 (Original submission)

Review form: Reviewer 1 (Marc Brysbaert)

Is the manuscript scientifically sound in its present form?

Yes

Are the interpretations and conclusions justified by the results?

Yes

Is the language acceptable?

Yes

Is it clear how to access all supporting data?

Yes

Do you have any ethical concerns with this paper?

No

Have you any concerns about statistical analyses in this paper?

No

Recommendation?

Accept with minor revision (please list in comments)

Comments to the Author(s)

Review ms Mollica & Piantadosi

Marc Brysbaert

This is an interesting ms estimating how much information is needed to acquire a language. It follows similar attempts to estimate the size of human memory.

To some extent, reviewing this ms could require weeks of work, properly checking each and every assumption / choice the authors make, to make sure no errors are present in the calculations. Unfortunately, I do not have the time (nor the motivation) to do so.

Instead, I've read the ms as an informed reader, judging whether the authors' assumptions / calculations agreed with mine for the areas I know best. I am happy to say that I am satisfied in this respect. This does not preclude that other researchers may take up the challenge and come up with different numbers based on different arguments. However, the beauty of the present submission is that the choices are made explicit and are open to challenge. As such, it presents a sound basis for scientific discussion.

The ms is exceptionally well written. As it happens, I have only one issue. It is the estimate of word frequency at the bottom of p. 7. Using a factor of two to get the upper limit would give a limit of 4 bits rather than 3, no? Wouldn't change much in terms of conclusions, but in my understanding would be more correct.

I also like the estimate of information required for syntax relative to lexical information. Indeed shines new light on the nativist/empirist discussion. Arguably will also be like by researchers of embodied cognition, as it shows the challenge of learning new words and word meanings.

Review form: Reviewer 2 (Dale Barr)**Is the manuscript scientifically sound in its present form?**

Yes

Are the interpretations and conclusions justified by the results?

Yes

Is the language acceptable?

Yes

Is it clear how to access all supporting data?

Yes

Do you have any ethical concerns with this paper?

No

Have you any concerns about statistical analyses in this paper?

No

Recommendation?

Accept as is

Comments to the Author(s)

I congratulate the authors on this provocative and bold study. Researchers have adduced large amounts of knowledge about language acquisition and representation, and the authors do an admirable job of synthesizing existing facts and estimates and show impressive creativity in generating further plausible-sounding estimates from existing sources when required. The result is a range of estimates of the amount of information that must be acquired to become a competent adult speaker of a language. I do not expect this to be anywhere near a definitive statement on this issue, but it is certain to spark interest and discussion, as well as alternative attempts to quantify the information encoded in the linguistic representations of a competent adult language user.

The claim of 1.5 MB is built on a fairly large pile of assumptions and back-of-the-envelope calculations, and so it is anyone's guess how close the authors are to the real target number. But at least the assumptions have been very transparently laid out in detail, and the authors are honest about the limitations of the study. Some of the math was beyond my expertise, but those equations that I could grasp seemed sound. The manuscript is also clearly written and practically flawless. I much appreciated the vividly written discussion section ("our knowledge of language fits compactly on a floppy disk"), which ends a provocative note ("any innate capacity for human language learning is most likely to be for lexical semantics"; probably too provocative, since learners probably spend most of the assumed 18 years learning about lexical semantics anyway).

Others will be better placed than I to evaluate the linguistic assumptions made at each of the representational levels. I have no doubt that linguistic experts will find much to debate regarding many assumptions behind these calculations, but you have to start somewhere. I would also guess that there would be some controversy about what types of representations should be considered part and parcel of Language. For instance, conventions of language use such as described by conversation analysts or Herbert Clark are not included, while word frequency is, and some might see word frequency as a "performance" aspect of language and not part of language knowledge proper. But I don't think these possible points of disagreement are really that important---what I think is important is the existence proof that the field has matured enough that such theoretical calculations are within our grasp, and that the authors have worked transparently so that others can evaluate and build on their work. Anyone who disagreed on the details could follow the authors' logic and generate their own estimates.

In sum, I think this paper makes a valuable contribution to the literature. I hope that publication is not held up by disagreements about the authors' assumptions (which are necessarily arbitrary in some cases, but I do think they more or less represent reasonable, best guess attempts for each of the relevant levels of representation). In my view, the main contribution here is the novel theoretical approach taken by the authors.

Decision letter (RSOS-181393.R0)

11-Feb-2019

Dear Dr Mollica

On behalf of the Editors, I am pleased to inform you that your Manuscript RSOS-181393 entitled "Humans store ~1.5 megabytes during language acquisition: information theoretic bounds" has been accepted for publication in Royal Society Open Science subject to minor revision in accordance with the referee suggestions. Please find the referees' comments at the end of this email.

The reviewers and handling editors have recommended publication, but also suggest some minor revisions to your manuscript. Therefore, I invite you to respond to the comments and revise your manuscript.

- Ethics statement

- Data accessibility

<http://datadryad.org/submit?journalID=RSOS&manu=RSOS-181393>

- Competing interests

- Authors' contributions

- Acknowledgements

- Funding statement

Because the schedule for publication is very tight, it is a condition of publication that you submit the revised version of your manuscript before 20-Feb-2019. Please note that the revision deadline will expire at 00.00am on this date. If you do not think you will be able to meet this date please let me know immediately.

- 1) A text file of the manuscript (tex, txt, rtf, docx or doc), references, tables (including captions) and figure captions. Do not upload a PDF as your "Main Document";
- 2) A separate electronic file of each figure (EPS or print-quality PDF preferred (either format should be produced directly from original creation package), or original software format);
- 3) Included a 100 word media summary of your paper when requested at submission. Please ensure you have entered correct contact details (email, institution and telephone) in your user account;
- 4) Included the raw data to support the claims made in your paper. You can either include your data as electronic supplementary material or upload to a repository and include the relevant doi

within your manuscript. Make sure it is clear in your data accessibility statement how the data can be accessed;

5) All supplementary materials accompanying an accepted article will be treated as in their final form. Note that the Royal Society will neither edit nor typeset supplementary material and it will be hosted as provided. Please ensure that the supplementary material includes the paper details where possible (authors, article title, journal name).

on behalf of Dr Shirley-Ann Rüschemeyer (Associate Editor) and Antonia Hamilton (Subject Editor)
openscience@royalsociety.org

Associate Editor Comments to Author (Dr Shirley-Ann Rüschemeyer):

Dear Dr. Mollica,

Thank you for considering Royal Society Open Science as an outlet for your work. Two reviewers and I have read your manuscript, and we are all of the opinion that it provides an interesting argument and a provocative platform for more debate. I am happy to recommend that the manuscript be accepted as it is. If you would like to make the adjustment that Reviewer 1 highlights, you are welcome to, but I leave that decision up to you.

With best wishes,

Reviewer comments to Author:

Reviewer: 1

Comments to the Author(s)

Review ms Mollica & Piantadosi

Marc Brysbaert

This is an interesting ms estimating how much information is needed to acquire a language. It follows similar attempts to estimate the size of human memory.

To some extent, reviewing this ms could require weeks of work, properly checking each and every assumption / choice the authors make, to make sure no errors are present in the calculations. Unfortunately, I do not have the time (nor the motivation) to do so.

Instead, I've read the ms as an informed reader, judging whether the authors' assumptions / calculations agreed with mine for the areas I know best. I am happy to say that I am satisfied in this respect. This does not preclude that other researchers may take up the challenge and come up with different numbers based on different arguments. However, the beauty of the present submission is that the choices are made explicit and are open to challenge. As such, it presents a sound basis for scientific discussion.

The ms is exceptionally well written. As it happens, I have only one issue. It is the estimate of word frequency at the bottom of p. 7. Using a factor of two to get the upper limit would give a limit of 4 bits rather than 3, no? Wouldn't change much in terms of conclusions, but in my understanding would be more correct.

I also like the estimate of information required for syntax relative to lexical information. Indeed shines new light on the nativist/empirist discussion. Arguably will also be like by researchers of embodied cognition, as it shows the challenge of learning new words and word meanings.

Reviewer: 2

Comments to the Author(s)

I congratulate the authors on this provocative and bold study. Researchers have adduced large amounts of knowledge about language acquisition and representation, and the authors do an admirable job of synthesizing existing facts and estimates and show impressive creativity in generating further plausible-sounding estimates from existing sources when required. The result is a range of estimates of the amount of information that must be acquired to become a competent adult speaker of a language. I do not expect this to be anywhere near a definitive statement on this issue, but it is certain to spark interest and discussion, as well as alternative attempts to quantify the information encoded in the linguistic representations of a competent adult language user.

The claim of 1.5 MB is built on a fairly large pile of assumptions and back-of-the-envelope calculations, and so it is anyone's guess how close the authors are to the real target number. But at least the assumptions have been very transparently laid out in detail, and the authors are honest about the limitations of the study. Some of the math was beyond my expertise, but those equations that I could grasp seemed sound. The manuscript is also clearly written and practically

flawless. I much appreciated the vividly written discussion section ("our knowledge of language fits compactly on a floppy disk"), which ends a provocative note ("any innate capacity for human language learning is most likely to be for lexical semantics"; probably too provocative, since learners probably spend most of the assumed 18 years learning about lexical semantics anyway).

Others will be better placed than I to evaluate the linguistic assumptions made at each of the representational levels. I have no doubt that linguistic experts will find much to debate regarding many assumptions behind these calculations, but you have to start somewhere. I would also guess that there would be some controversy about what types of representations should be considered part and parcel of Language. For instance, conventions of language use such as described by conversation analysts or Herbert Clark are not included, while word frequency is, and some might see word frequency as a "performance" aspect of language and not part of language knowledge proper. But I don't think these possible points of disagreement are really that important---what I think is important is the existence proof that the field has matured enough that such theoretical calculations are within our grasp, and that the authors have worked transparently so that others can evaluate and build on their work. Anyone who disagreed on the details could follow the authors' logic and generate their own estimates.

In sum, I think this paper makes a valuable contribution to the literature. I hope that publication is not held up by disagreements about the authors' assumptions (which are necessarily arbitrary in some cases, but I do think they more or less represent reasonable, best guess attempts for each of the relevant levels of representation). In my view, the main contribution here is the novel theoretical approach taken by the authors.

Author's Response to Decision Letter for (RSOS-181393.R0)

See Appendix A.

Decision letter (RSOS-181393.R1)

25-Feb-2019

Dear Dr Mollica,

I am pleased to inform you that your manuscript entitled "Humans store ~1.5 megabytes during language acquisition: information theoretic bounds" is now accepted for publication in Royal Society Open Science.

Royal Society Open Science operates under a continuous publication model (<http://bit.ly/cpFAQ>). Your article will be published straight into the next open issue and this will be the final version of the paper. As such, it can be cited immediately by other researchers.

As the issue version of your paper will be the only version to be published I would advise you to check your proofs thoroughly as changes cannot be made once the paper is published.

on behalf of Dr Shirley-Ann Rüschemeyer (Associate Editor) and Professor Antonia Hamilton (Subject Editor)
openscience@royalsociety.org

Appendix A

Dear Editors,

Thank you for great news! We were real pleased with the reviewers' comments and are happy to submit the manuscript in the requested format.

Thank you for considering Royal Society Open Science as an outlet for your work. Two reviewers and I have read your manuscript, and we are all of the opinion that it provides an interesting argument and a provocative platform for more debate. I am happy to recommend that the manuscript be accepted as it is. If you would like to make the adjustment that Reviewer 1 highlights, you are welcome to, but I leave that decision up to you.

We have slightly changed the abstract and last paragraph to clarify/simplify our position. With regard to Review 1, we did not change the manuscript but respond below.

Thank you.

Best,
Frank Mollica & Steve Piantadosi

The ms is exceptionally well written. As it happens, I have only one issue. It is the estimate of word frequency at the bottom of p. 7. Using a factor of two to get the upper limit would give a limit of 4 bits rather than 3, no? Wouldn't change much in terms of conclusions, but in my understanding would be more correct.

We introduce the factor of error on the estimate of the mental representations of the bins not the bit estimates themselves. So our best guess 4 bins = $\log_2(4) = 2$ bits per word. Lower bounded as 2 bins = $\log_2(2) = 1$ bit per word. Upper bounded as 8 bins = $\log_2(8) = 3$ bits per word.